# Gradeability of a Forwarder Based on Traction Performance

Zoran Bumber [1], Andreja Đuka [2,*], Zdravko Pandur [2] and Tomislav Poršinsky [2]

[1] Croatian Forests Ltd., Forest Administration Zagreb, 10000 Zagreb, Croatia
[2] Department of Forest Engineering, Faculty of Forestry and Wood Technology, University of Zagreb, 10000 Zagreb, Croatia
* Correspondence: aduka@sumfak.unizg.hr

**Abstract:** Based on the knowledge of the dimensional and mass features of a forwarder, a model was developed to assess its mobility during timber forwarding uphill in a safe and eco-efficient way. The model is based on knowledge of the position of the forwarder's centre of gravity, its declared payload and the length of the loaded timber, as well as the gradeability for uphill timber forwarding based on the traction characteristics of the vehicle. The model connects two research approaches, (1) vehicle–terrain approach (distribution of axle loads depending on the longitudinal terrain slope) and (2) wheel–soil approach (estimation of the traction characteristics of the forwarder based on the wheel numeric), concerning previous research: (i) underload on the front axle of the vehicle, (ii) overload on the rear axle of the vehicle, (iii) permissible tire load, (iv) minimal soil bearing capacity, (v) wheel slip. Simulation modelling for the assessment of the forwarders' mobility range during timber forwarding uphill was conducted on an example of an eight-wheel Komatsu 875 forwarder, with a declared payload of 16,000 kg, equipped with 710/45-26.5 tires, for which the position of the centre of gravity was determined by the method of lifting the axle. The results of the distribution of the adhesion load on the front and rear axles of the forwarder indicated that, during timber forwarding of 16,000 kg and 4.82 m long hardwood logs on a terrain slope below 68%, there is no critical unloading on the front bogie axle, nor overloading on the rear bogie axle, i.e., wheel tire overload that could limit forwarder mobility. For the specified range of longitudinal terrain slope, a minimal cone index of 950 kPa for an exemplary forwarder is an environmental factor and was calculated based on the nominal ground pressure of the reference (heavier loaded) rear wheels of the vehicle. The forwarders' mobility range was determined by the intersection curves of the gradeability (based on forwarders' traction characteristics at wheel slip of 25% vs. cone index) and the curve of the minimal soil cone index.

**Keywords:** forwarders' centre of gravity; declared payload; axle load distribution; wheel numeric

## 1. Introduction

When evaluating the applicability of forest vehicles for timber felling, processing and extraction in an effective, safe and environmentally acceptable way, forestry experts are required to know the mobility of forest vehicles [1–3]. The mobility of forest vehicles is their ability to travel from point A to point B in a forest stand (cut-block) while maintaining: (1) the ability to perform their primary task and (2) maintain environmental suitability [4]. The presence and level of terrain factors (terrain slope, ground obstacles and soil bearing capacity) determine terrain trafficability, which eventually enables or prevents the mobility of forest vehicles [5]. Considering the complexity of the interaction between vehicles and the terrain, the mobility of forest vehicles is divided into (1) manoeuvrability—the ability to overcome terrain irregularities, during which interaction of two geometric systems occurs, i.e., the geometry of the vehicle and the geometry of the terrain surface [6]—(2) traction performance—dependence of traction force on wheel slip and soil bearing capacity [7,8]—and (3) environmental suitability—vehicles' ground contact pressures [9,10].

The mobility of forwarders is becoming a research challenge due to their increased use on sloped terrain [11–13]. Due to winch-assist forwarders, timber forwarding is becoming cost-competitive to skyline logging [14,15]. In addition to timber forwarding distance [16,17], the most significant influencing factor of productivity and timber unit costs in timber transport is timber volume (mass) transported in each cycle [18,19]. Based on the knowledge of the dimensional and mass features of a forwarder, Weise [20–22] developed a load distribution plan on a forwarder on a horizontal surface intending to determine the mass (but also the length) of the loaded roundwood of hardwood and softwood while respecting four restrictions, whereby the load mass: (1) must not be greater than the vehicle's declared payload by the manufacturer and (2) does not overload the front or rear axle of the vehicle (that is, the sum of the tire load capacity per axle) but also (3) does not relieve the front axle of the vehicle.

Nominal ground pressure [23] is a generally accepted way of defining contact pressures of forest vehicles on forest soil [24,25] and a base for determining minimum soil bearing capacity [26] as a guideline for environmentally sound timber extraction. Hittenbeck [8], in researching forwarder traction on sloped terrain and soils with different water content and share of stoniness, concludes that a wheel slip of 25% represents an environmental limitation of the application of a forwarder to prevent future erosion processes.

The most common question when choosing a forwarder and planning timber forwarding is: which slope (and direction) of the terrain and which soil bearing capacity are sufficient for timber forwarding of the officially declared/allowed payload by the vehicle manufacturer? The answer to the question is not simple. Measurements of the entire series of forces, resistances and torque are a long and expensive procedure. Obtained measurement results are often applicable only to the conditions in which the measurements were made, depending on the type of a forwarder and its various equipment i.e., tires, chains and tracks, terrain slope, the direction of timber forwarding i.e., downhill vs. uphill, soil water content and bearing capacity, and mass and length of loaded timber. Simulation modelling gives a solution by connecting two research approaches in assessing the mobility of forwarders during timber forwarding (Figure 1), with which it is possible to cover a broader range of values of the mentioned influential factors while respecting the limitations that arose from previous research.

The theoretical approach for the distribution of forces during timber forwarding uphill is shown in Figure 1A, where the forces are divided into vertical, horizontal and traction forces, which is, in the literature, referred to as the "vehicle–terrain system". Resistance forces of the forest tractor are often divided into five groups [27,28]: (1) slope resistance, (2) drawbar pull, (3) rolling resistance, (4) air resistance, and (5) inertia resistance.

Drawbar pull is zero for forwarders because forwarders only overcome rolling resistance during movement [7,29]. When the velocity is constant, inertia resistance is zero. Air resistance becomes negligible at the low velocities attainable on the forest floor. In practice, when forwarding timber uphill [8], only rolling resistance and slope resistance are significant (Equation (1)). Dividing Equation (1) with the adhesive weight of the nominally loaded forwarder results in an equation for the gross traction factor for timber forwarding (Equation (2)), which shows how much of the adhesive weight of the forwarder is converted into thrust force.

$$F_t = (G + G_{\text{load}}) \cdot \cos\alpha \cdot f + (G + G_{\text{load}}) \cdot \sin\alpha = (G + G_{\text{load}}) \cdot (\cos\alpha \cdot f + \sin\alpha) \quad (1)$$

$$\kappa = \frac{(G + G_{\text{load}}) \cdot (\cos\alpha \cdot f + \sin\alpha)}{(G + G_{\text{load}}) \cdot \cos\alpha} = \frac{\cos\alpha \cdot f + \sin\alpha}{\cos\alpha} = f + tg\alpha \quad (2)$$

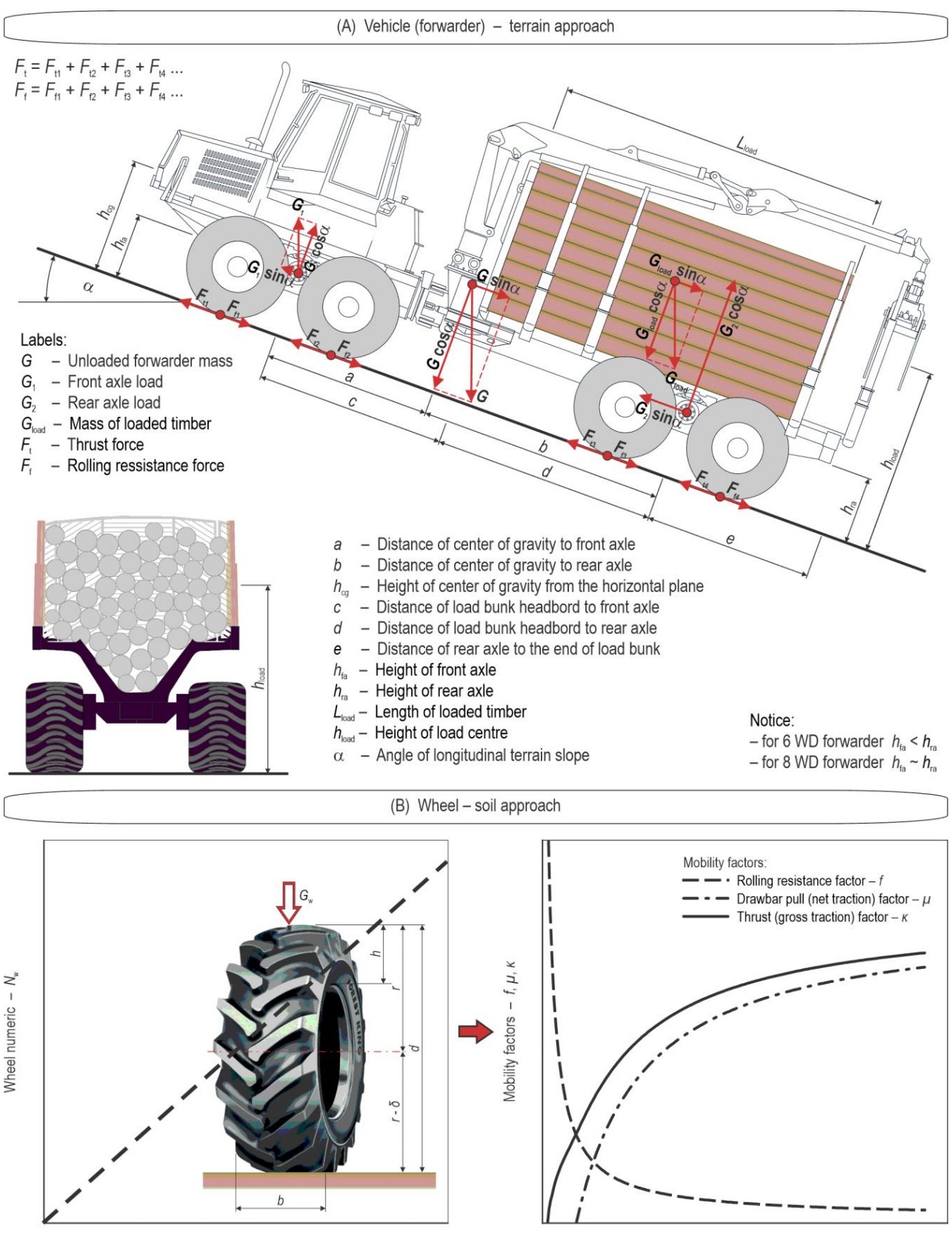

**Figure 1.** Research approaches to forwarder mobility.

The descriptions of the symbols in Equations (1) and (2) are presented in Figure 1.

The mobility of the vehicle also depends on its traction performance, i.e., the dependence of the thrust/traction force on wheel slip and soil bearing capacity [30,31]. By developing an empirical method of researching a complex wheel–soil system (Figure 1B), known in the literature as the WES method (Waterways Experimentation Station, U.S. Army Corps of Engineering Research), it is possible to connect the traction performance of the vehicle, soil bearing capacity (cone index) to wheel numeric [29]. The soil-bearing capacity of forest soil is usually determined by the penetration of a cone into the ground and is defined as the ratio of force required to press the standardised cone, as well as varying grounds' resistance to penetration depending on its depth. The ground penetration curve will therefore contain data on the estimation of soil strength depending on the depth of the cone penetration, caused by the horizon condition of certain soil types [29]. The wheel numeric is a dimensionless parameter (factor) that describes the interaction between the loaded wheel and the soil. This factor is determined by the ratio of the nominal ground pressure and the soil bearing capacity determined with a penetrometer [32]. With gross and net traction factors and rolling resistance factors, it is possible to determine the traction performance of the vehicle (thrust and drawbar pull, rolling resistance) based on the vehicle's wheel load (Figure 1B). The area above the curve of the gross traction factor is the area of the impossible vehicle movement because, in that case, the resistance forces are greater than the thrust force [32]. In the case of different dimensions of the front and rear wheels of the vehicle, unequal load distribution between the front and rear axles occurs. The so-called reference wheel is used to assess vehicle mobility according to the WES method. The reference wheel is the wheel with the lowest value of the wheel numeric, i.e., the highest load [29].

The goals of this paper are: (i) to measure the position of the centre of gravity of an exemplary Komatsu 875 eight-wheeled forwarder, (ii) to develop a model of the axle load distribution of a nominally loaded forwarder during timber forwarding uphill, (iii) to determine the nominal ground pressure and the minimum soil cone index, (iv) to use simulation modelling to determine the gradeability of a forwarder depending on the soil cone index and wheel slip, (v) to determine the mobility range of the forwarder for the future planning of timber forwarding uphill in a safe, effective and environmentally acceptable way.

## 2. Materials and Methods

### 2.1. Komatsu 875 Forwarder

The model for assessing the mobility of the forwarder for timber forwarding uphill is based on the Komatsu 875 forwarder, with a declared payload of 16,000 kg (Figure 2). The angle of articulation of the forwarder is $\pm42°$, and the vertical mobility of the forwarder when driving on terrain is ensured by the frame oscillation of $\pm16°$, as well as the bogie axle assembly wheelbase angle at $\pm20°$. The vehicle is powered by a six-cylinder precharged diesel engine (AGCO Power 74-AWF) with a displacement of 7400 $cm^3$, a maximum power of 190 kW at 1900 $min^{-1}$ and a torque of 1130 Nm at 1500 $min^{-1}$. The forwarder is equipped with a Komatsu 145F hydraulic crane weighing 2400 kg and with a reach of 8.5 m, with a gross lifting torque of 145 kNm and a gross slewing torque of 38 kNm.

The front and rear bogie axles of the Komatsu 875 forwarder are the same—NAF PTA 76, for which the manufacturer states a maximum static load of 360 kN and a maximum dynamic load of 290 kN [33]. The wheels of the front and rear bogie axles are equipped with tires of the same dimensions 710/45-26.5 20 PR (Nokian Tyres Forest King TRS 2), whose load capacity at an air inflation pressure of 500 kPa and a speed < 10 km/h is 6.9 t/tire [34].

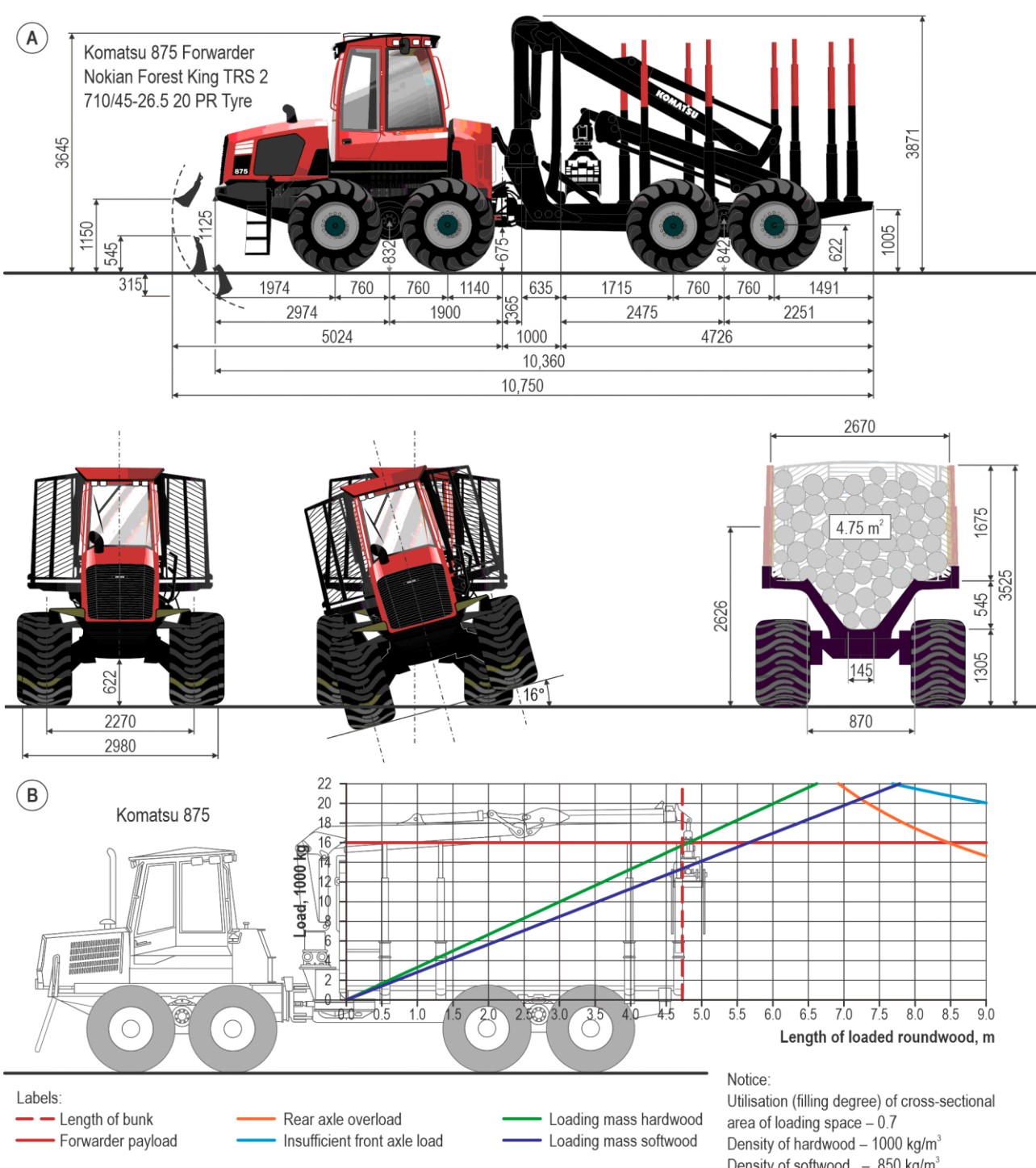

**Figure 2.** Dimensions (**A**) and load distribution plan (**B**) of Komatsu 875 forwarder [35].

The cross-section of the loading area is 4.75 m², and the length is 4.726 m. Poršinsky et al. [35], showing the dimensional and mass features of the Komatsu 875 forwarder following the ISO 13860 [36] standard, also created a load distribution plan for this forwarder for hardwood (deciduous) and softwood (coniferous) timber according to KWF's (Kuratorium für Waldarbeit und Forsttechnik) methodology [20–22]. The authors concluded that, when the cross-section of the loading area is filled with timber of hardwood 4.82 m long and softwood 5.65 m long, there will be no: (1) exceeding the declared payload by the manufacturer, (2) front and rear axle load capacity overloads, (3) tire overloads, and (4) insufficient front axle load.

### 2.2. Determining the Centre of Gravity of the Forwarder

The forwarder's position of the centre of gravity will be determined by the lifting axle method [37]. Weighing an unloaded forwarder on a flat surface will determine the loads on individual wheels of the vehicle (by using portable 10-ton Telub scales) i.e., its axle loads, which is necessary for determining the horizontal distance of the centre of gravity from the front and rear axles of the forwarder (Figure 3A). By knowing the forwarder wheelbase and respecting the conditions shown in Equations (3) and (4) and setting the equilibrium equation around the front axle (Equation (5)) of the forwarder (positive momentums are in the clockwise direction), equations were derived for calculating the horizontal distances of the centre of gravity from the front (Equation (6)) and rear axles (Equation (7)).

$$G_1 + G_2 = G \tag{3}$$

$$a + b = L \tag{4}$$

$$G \cdot a - G_2 \cdot L = 0 \tag{5}$$

$$a = \frac{G_2 \cdot L}{G} \tag{6}$$

$$b = L - a \tag{7}$$

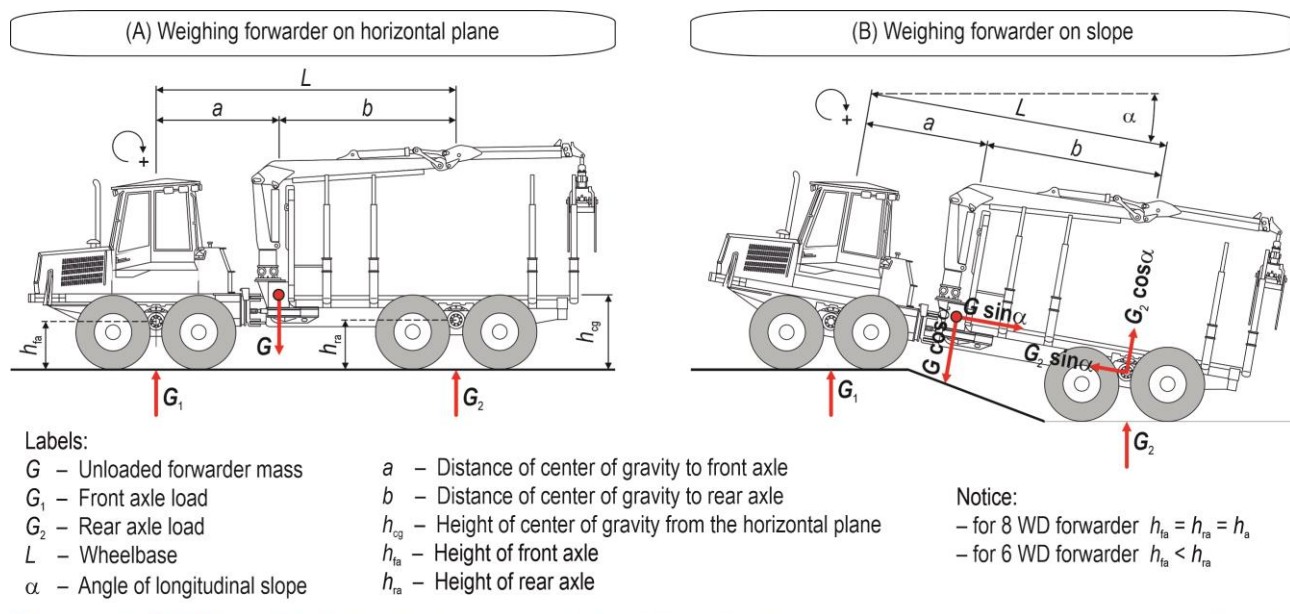

Labels:
$G$ – Unloaded forwarder mass
$G_1$ – Front axle load
$G_2$ – Rear axle load
$L$ – Wheelbase
$\alpha$ – Angle of longitudinal slope

$a$ – Distance of center of gravity to front axle
$b$ – Distance of center of gravity to rear axle
$h_{cg}$ – Height of center of gravity from the horizontal plane
$h_{fa}$ – Height of front axle
$h_{ra}$ – Height of rear axle

Notice:
– for 8 WD forwarder $h_{fa} = h_{ra} = h_a$
– for 6 WD forwarder $h_{fa} < h_{ra}$

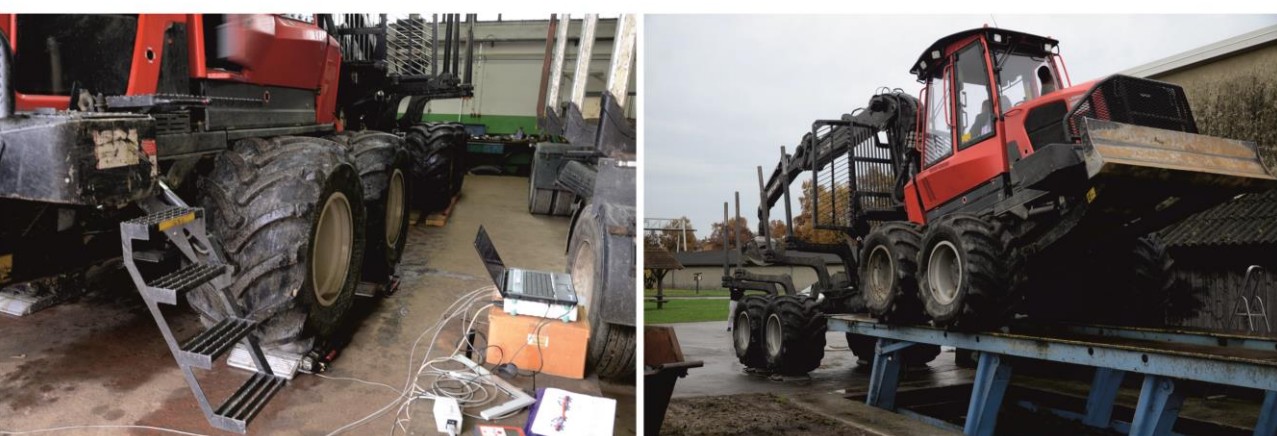

**Figure 3.** Determining the centre of gravity of the forwarder in (**A**) horizontal plane and (**B**) on slope.

The descriptions of the symbols are presented in Figure 3.

Weighing the load under the rear wheels of an unloaded forwarder on a slope (a 25-ton lifting platform was used) was carried out to determine the change in load on the rear axle of the forwarder concerning the known longitudinal slope of the vehicle (Figure 3B). By setting the equilibrium equation around the front axle (Equation (8)) of the forwarder (positive momentums are clockwise), an equation for calculating the height of the centre of gravity of the unloaded forwarder was derived (Equation (9)). Given that the axles and tires of the eight-wheeled forwarders are of the exact dimensions, and ignoring the different wheel deflections of the forwarder's front and rear axles, it follows that the heights of the forwarder's front and rear axles are the same ($h_{fa} = h_{ra} = h_a$). Thus, the equation for calculating the height of the centre of gravity of the eight-wheeled forwarder takes the form shown in Equation (10).

$$G \cdot \cos \alpha \cdot a + G \cdot \sin \alpha \cdot \left( h_{cg} - h_{fa} \right) - G_2 \cdot \cos \alpha \cdot L + G_2 \cdot \sin \alpha \cdot \left( h_{ra} - h_{fa} \right) - G_1 \cdot 0 = 0 \quad (8)$$

$$h_{cg} = \frac{G_2 \cdot \cos \alpha \cdot L - G \cdot \cos \alpha \cdot a - G_2 \cdot \sin \alpha \cdot \left( h_{ra} - h_{fa} \right)}{G \cdot \sin \alpha} + h_{fa} \quad (9)$$

$$h_{cg} = \frac{G_2 \cdot \cos \alpha \cdot L - G \cdot \cos \alpha \cdot a}{G \cdot \sin \alpha} + h_a \quad (10)$$

The descriptions of the symbols are presented in Figure 3.

### 2.3. Forwarder Axle Load Distribution Model during Timber Forwarding Uphill

The forwarder axle load distribution model is based on the position of the centre of gravity and other dimensional and mass features of the forwarder shown in Figure 1A, as well as the mass of loaded timber at a full cross-sectional area of loading space, same as in German KWF test reports (Equation (11)). KWF suggests using a value of 0.7 for the full cross-sectional area of loading space i.e., a wood density of 1000 kg/m$^3$ (hardwood) and 700 kg/m$^3$ (softwood).

$$G_{\text{load}} = A \cdot f \cdot \rho \cdot s \quad (11)$$

where:

$G_{\text{load}}$—mass of loaded timber (kg)
$A$—cross-sectional area of loading space (m$^2$)
$f$—utilisation (filling degree) of the cross-sectional area of loading space
$\rho$—wood density of loaded timber (kg/m$^3$)
$s$—length of loaded timber (m).

The load on the front axle of the forwarder is derived by setting the equilibrium equation around the rear axle (positive momentums are counter-clockwise)—Equation (12), that is, by arranging Equation (13).

$$G \cdot \cos \alpha \cdot b - G \cdot \sin \alpha \cdot \left( h_{cg} - h_{ra} \right) + G_{\text{load}} \cdot \cos \alpha \cdot \left( d - \tfrac{L_{\text{load}}}{2} \right) - G_{\text{load}} \cdot \sin \alpha \cdot \left( h_{\text{load}} - h_{ra} \right) - G_1 \cdot \cos \alpha \cdot (a+b) - $$
$$-G_1 \cdot \sin \alpha \cdot \left( h_{fa} - h_{ra} \right) = 0 \quad (12)$$

$$G_1 = \frac{G \cdot \left[ \cos \alpha \cdot b - \sin \alpha \cdot \left( h_{cg} - h_{ra} \right) \right] + G_{\text{load}} \cdot \left[ \cos \alpha \cdot \left( d - \tfrac{L_{\text{load}}}{2} \right) - \sin \alpha \cdot \left( h_{\text{load}} - h_{ra} \right) \right]}{\cos \alpha \cdot (a+b) + \sin \alpha \cdot \left( h_{fa} - h_{ra} \right)} \quad (13)$$

The forwarder's load on the rear axle is derived by setting the equilibrium equation around the front axle (positive momentums are clockwise)—Equation (12), that is, by arranging Equation (13).

$$G \cdot \cos \alpha \cdot a + G \cdot \sin \alpha \cdot \left( h_{cg} - h_{fa} \right) + G_{\text{load}} \cdot \cos \alpha \cdot \left( \tfrac{L_{\text{load}}}{2} + c \right) + G_{\text{load}} \cdot \sin \alpha \cdot \left( h_{\text{load}} - h_{fa} \right) - G_2 \cdot \cos \alpha \cdot (a+b) - $$
$$-G_2 \cdot \sin \alpha \cdot \left( h_{ra} - h_{fa} \right) = 0 \quad (14)$$

$$G_2 = \frac{G \cdot \left[ \cos\alpha \cdot a + \sin\alpha \cdot \left( h_{cg} - h_{fa} \right) \right] + G_{\text{load}} \cdot \left[ \cos\alpha \cdot \left( \frac{L_{\text{load}}}{2} + c \right) + \sin\alpha \cdot \left( h_{\text{load}} - h_{fa} \right) \right]}{\cos\alpha \cdot (a + b) + \sin\alpha \cdot \left( h_{ra} - h_{fa} \right)} \tag{15}$$

The descriptions of the symbols are presented in Figures 1 and 3.

The height of the load's centre of gravity in Equations (13) and (15) was calculated under the ISO 13860 [36] standard, while the other dimensional features of the exemplary forwarder are shown in Figure 2.

### 2.4. Nominal Ground Pressure and Minimal Cone Index

The forwarder wheel load ($G_w$), depending on the longitudinal slope of the terrain ($\alpha$) and the declared quantity (mass) of loaded timber, will presumably have an equal distribution of the adhesive axle load. Based on the forwarder wheel load, the reference wheel of the vehicle (as the one with the highest load) will be determined, but also the nominal ground pressure of the forwarder will be calculated [23] according to Equation (16), i.e., the value of the minimum soil bearing capacity [26] based on guidelines for environmentally acceptable timber harvesting (Equation (17)).

$$NGP = \frac{G_{\mathbf{w}}}{r \cdot b} \tag{16}$$

$$CI_{\min} = 7.2 \cdot NGP \tag{17}$$

### 2.5. Forwarder Traction

The evaluation of the traction performance of the Komatsu 875 forwarder will be determined based on the Briuxius model [32]. The model is based on: (1) the load of the vehicles' reference wheel, (2) soil cone index, (3) wheel slip at 10%–15%, which corresponds to the highest tractive efficiency of the vehicle, and at 20%–25%, which corresponds to the limiting value of environmental acceptability [8]. Brixius's model of vehicle mobility is based on equations: (1) wheel numeric (Equation (18)), (2) rolling resistance factor (Equation (19)), and (3) gross traction factor (Equation (20)).

$$N_{\mathbf{w}} = \left( \frac{CI \cdot b \cdot d}{G_{\mathbf{w}}} \right) \left( \frac{1 + 5\frac{\delta}{h}}{1 + 3\frac{b}{d}} \right) \tag{18}$$

$$f = \frac{1.0}{N_{\mathbf{w}}} + 0.04 + \frac{0.5 \cdot \delta}{\sqrt{N_{\mathbf{w}}}} \tag{19}$$

$$\kappa = 0.88 \left( 1 - e^{-0.1\, N_{\mathbf{w}}} \right) \left( 1 - e^{-7.5\, \delta} \right) \tag{20}$$

$$\delta = 0.008 + 0.01 \left[ 0.365 + \left( \frac{170}{p_{\mathbf{i}}} \right) \right] \cdot G_{\mathbf{w}} \tag{21}$$

The mentioned equations that predict the values of the wheel numeric and the gross traction and resistance factors were developed by a regression analysis of the measurement results of 121 experiments, i.e., the combination of soil and tire dimensions, i.e., wheel load. Given that the wheel numeric is also based on the tire deflection, an empirical equation [29] was used to estimate this parameter based on tire pressure and wheel load (Equation (21)). Although the Brixsius model was developed for the purpose of assessing the mobility of agricultural tractors on arable land, many authors [4,38–41] consider it the most suitable for assessing the mobility of forest vehicles.

### 2.6. Gradeability of Forwarder

The limit terrain slope in timber forwarding uphill is based on the traction performance based on the reference wheel load values of the nominally loaded forwarder at different

longitudinal terrain slopes. The values of tg $\alpha$ are equated with the difference of the gross traction factor (Equation (20)) and the rolling resistance factor (Equation (19)) calculated on the basis of wheel load distribution and the Brixsius model, with the aim of determining the value of the soil cone index (Equation (22)). In this way, each value of the reference wheel depending on the longitudinal terrain slope during timber forwarding by a nominally loaded forwarder is associated with the value of the soil cone index (Equation (18)).

$$tg\alpha = \left[0.88\left(1 - e^{-0.1\cdot N_{\mathbf{w}}}\right)\left(1 - e^{-7.5\cdot \delta}\right)\right] - \left[\frac{1.0}{N_{\mathbf{w}}} + 0.04 + \frac{0.5\cdot \delta}{\sqrt{N_{\mathbf{w}}}}\right] \Rightarrow CI \qquad (22)$$

## 3. Results

In ISO 13860 standard [36], the axle loads of an unloaded forwarder on a horizontal surface are not determined by the position of the hydraulic crane, i.e., whether it is in the position of driving or extended position. Thus, crane position is neglected as the most accurate input data for the calculation of the load distribution plan and the calculation of the distribution of axle loads of the forwarder during timber forwarding uphill. The results of weighing the determined wheel loads and axle loads for the unloaded forwarder Komatsu 875 with the crane in the driving position are shown in Figure 4A and with the crane in the extended position in Figure 4B. The deviation of the forwarder's net mass of 5.1 kg with these two weighing versions is negligible (the forwarder's net mass with the crane in the position of driving is 21,380.4 kg, and with the extended crane 21,385.5 kg). The load on the front axle of the forwarder with the crane in the position of driving is 13,312 kg (62.3% of the net mass), and the load on the rear axle is 8067.8 kg, i.e., 37.7% of the net mass. When weighing the forwarder with the crane in the extended position, there was a change in the distribution of the load on the axles, where the front axle was loaded with 12,718.2 kg (59.5%) and the rear axle with 8667.3 kg (40.5%). Regardless of the position of the hydraulic crane when weighing the forwarder, the sum of the total loads of all right wheels was 50.6% of the net vehicle mass, i.e., the sum of loads of all left wheels was 49.4%, which indicates that the centre of gravity of the forwarder is negligibly shifted from the longitudinal bisector of the vehicle to the right side of the vehicle.

Weighing the forwarder on the lifting platform with a slope of 11.6° (Figure 3B) increased the load under the rear wheels (Figure 4C) compared to measurements on a horizontal surface. Weighing of an unloaded forwarder on a slope of 11.6° revealed an increase in the load on the rear axle of the forwarder compared to the weighing on a horizontal surface from 8067.8 kg to 8500.2 kg, respectively (crane in the position of driving), or from 8667.3 kg to 9205.7 kg, respectively (crane in extended position).

Knowing the distance between the axles (5375 mm) and applying the results of measuring the axle loads and the total mass of the forwarder on a horizontal surface with the use of Equations (6) and (7), the distance of the vehicle's centre of gravity from the front and rear axles was calculated for both crane positions (Figure 4D). Based on the results of the measurement of the load on the vehicles' rear axle on a slope of 11.6° and the calculated values of the horizontal distance of the centre of gravity point from the front axle, the height of the centre of gravity of the Komatsu 875 forwarder was calculated using Equation (10) (Figure 4D) and considering both positions of the crane.

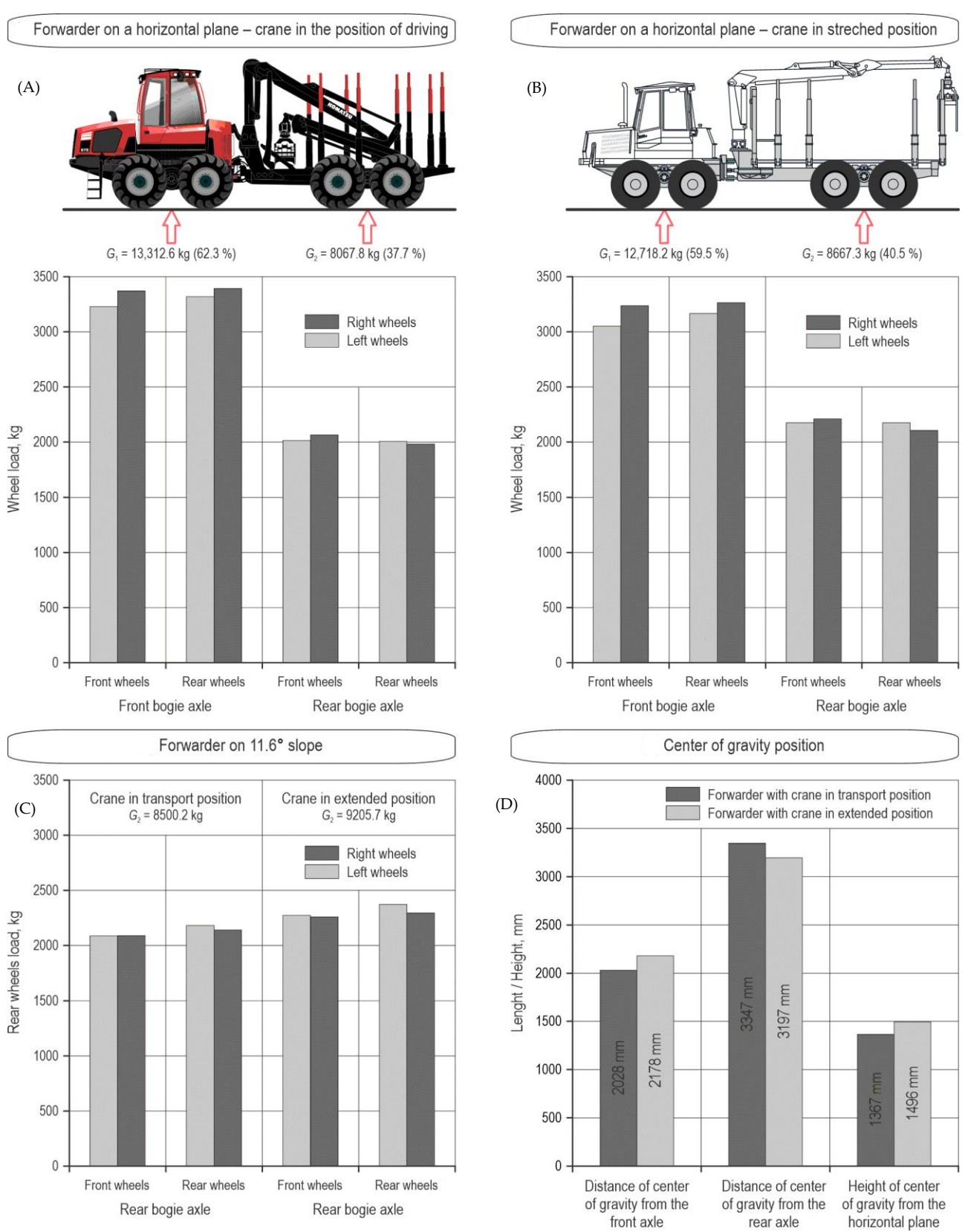

**Figure 4.** Wheel loads on a horizontal plane and slope and centre of gravity position—empty Komatsu 875.

The position of the centre of gravity of the Komatsu 875 forwarder, determined by the method of lifting the vehicle axle, indicated: (1) an increase in the distance of the centre of

gravity from the vehicle's front axle with the crane extended when compared to the crane in the driving position (2028 mm → 2178 mm), (2) a decrease in the distance of the forwarders' centre of gravity from the rear axle (3347 mm → 3197 mm), and (3) an increase in the height of the forwarder's centre of gravity (1367 mm → 1496 mm). The mentioned difference in the position of the centre of gravity is the result of moving part of the mass of the hydraulic crane's inner, outer, and telescope boom towards the rear axle of the vehicle, as well as positioning the crane in the plane of the stake height. To calculate the load on the vehicles' front axle (Equation (13)) and the rear axle (Equation (15)) of the nominally loaded Komatsu 875 forwarder depending on the longitudinal slope of the terrain, it is undoubtedly more convenient to use the position of the centre of gravity of the forwarder with the crane in extended position because it corresponds to the terrain working conditions. When the cross-sectional area of loading space (4.75 m$^2$) is loaded with hardwood (1000 kg/m$^3$) 4.82 m long, the declared payload (16 t) of the Komatsu 875 forwarder will not be exceeded (Equation (11)).

Figure 5 shows the distribution of axle and adhesion axle loads of the nominally loaded Komatsu 875 forwarder, depending on the longitudinal slope of the terrain < 70%.

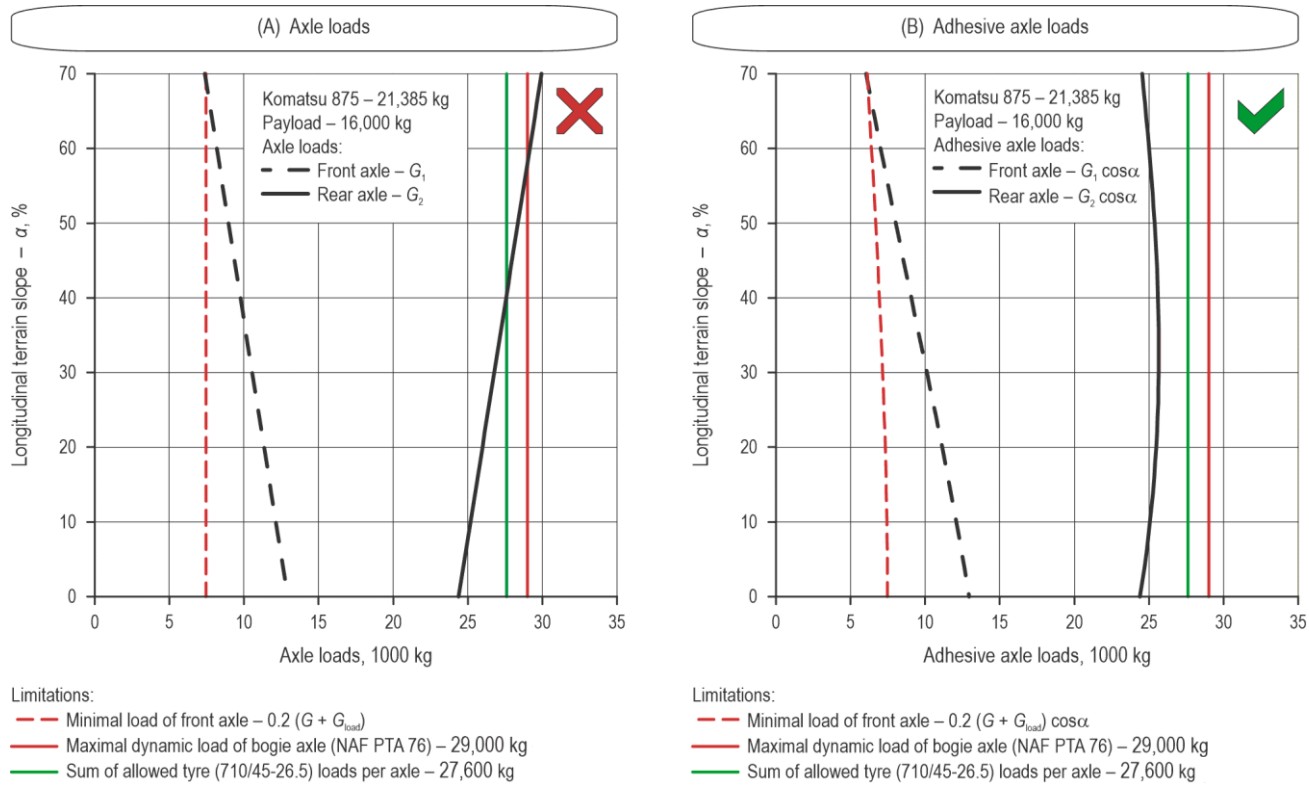

**Figure 5.** Axle loads and adhesive axle loads vs. terrain slope—Komatsu 875 forwarder [33,34].

For the specified range of longitudinal terrain slopes (Figure 5A), the load on the rear axle of the Komatsu 875 forwarder increases from 24.4 t to 30 t, while the load on the front axle decreases from 12.9 t (flat terrain) to 7.4 t (longitudinal slope of 70%). Critical points of the longitudinal terrain slope listed below should not be considered limitations of the forwarder's application because a gravity load is not perpendicular to the longitudinal terrain slope:

⇒　41%, at which the rear axle load reaches the sum of the allowed tire (710/45-26.5) loads per axle,

⇒　50%, at which the rear axle load reaches the permissible dynamic load of the NAF PTA 76 axle,

⇒　69% of the longitudinal terrain slope, after which the load on the front axle is below 20% of the total mass of the Komatsu 875 forwarder with the declared load.

The limitations of the forwarder application are related primarily to the adhesion load of the axles (Figure 5B), which is perpendicular to the longitudinal terrain slope. The adhesive load on the rear axle of the Komatsu 875 forwarder with nominal load ranges from 24.4 t (flat terrain) to 24.5 t (longitudinal terrain slope of 70%), with a maximum of 26.6 t on longitudinal terrain slopes between 30% and 40% without exceeding: (1) the sum of the allowed tire (710/45-26.5) loads per axle (27.6 t) and (2) the permissible dynamic load of bogie axles NAF PTA 76 (29 t). The decrease of the front axle load, which must not fall below 20% of the weight of the loaded forwarder, occurs at a longitudinal terrain slope of 68%.

Further analyses of calculating the nominal ground pressure, the wheel numeric, the gradeability, and the wheel load on the axles depending on the longitudinal terrain slope will be made with an assumption of an equal distribution of the adhesive axle load of the nominally loaded Komatsu 875 forwarder.

Figure 6 shows the distribution of the nominal ground pressure and the minimal cone index of the nominally loaded Komatsu 875 eight-wheeled forwarder equipped with 710/45-26.5 tires, depending on the longitudinal terrain slope up to 70%. The nominal ground pressure of the front bogie axle wheels ranges from 67 kPa (flat terrain), and by increasing the longitudinal terrain slope, it decreases due to the unloading of the front axle to 31 kPa (at a terrain slope of 70%). The nominal ground pressure of the wheels on the rear bogie axle ranges from 126 kPa (flat terrain), with the highest value of 132 kPa at longitudinal terrain slopes between 30% and 40% (Figure 6A). The environmental suitability of timber forwarding with the Komatsu 875 forwarder is shown by the analysis of the minimal cone index (Figure 6B), based on the values of the reference wheels of the rear bogie axle, which range from 906 kPa (flat terrain) to 950 kPa (longitudinal terrain slopes of 30% to 40%). The eco-efficient timber forwarding with the nominally loaded Komatsu 875 forwarder equipped with 710/45-26.5 tires is primarily related to favourable conditions of soil bearing capacity (strong soil).

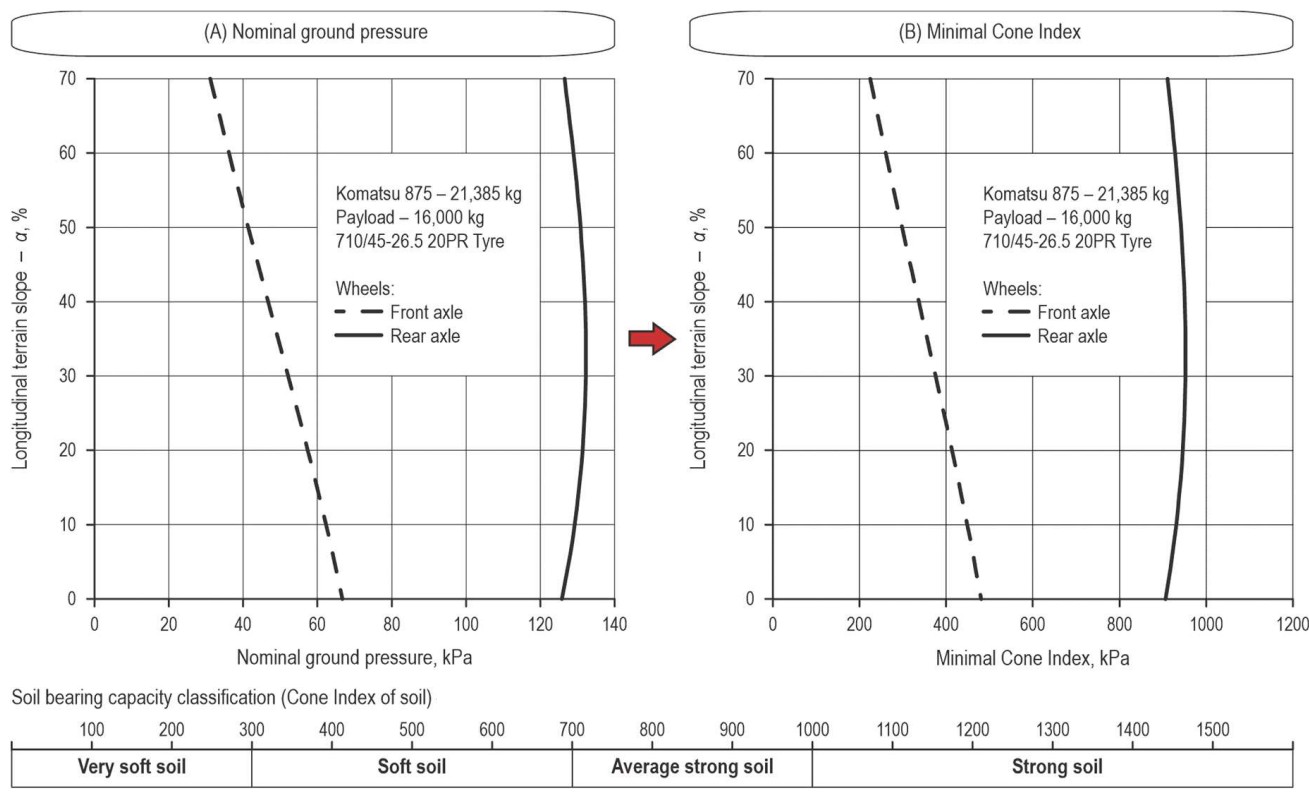

**Figure 6.** Nominal ground pressure and minimal cone index vs. terrain slope—Komatsu 875 forwarder [42].

Gradeability of the nominally loaded Komatsu 875 forwarder during uphill forwarding significantly depends on the soil bearing capacity expressed by the cone index and wheel slip (Figure 7A). At a constant cone index of 1 MPa, gradeability for forwarding is 20% at a wheel slip of 10%, 28% at a wheel slip of 15%, 32% at a wheel slip of 20%, and 36% at a wheel slip of 25%.

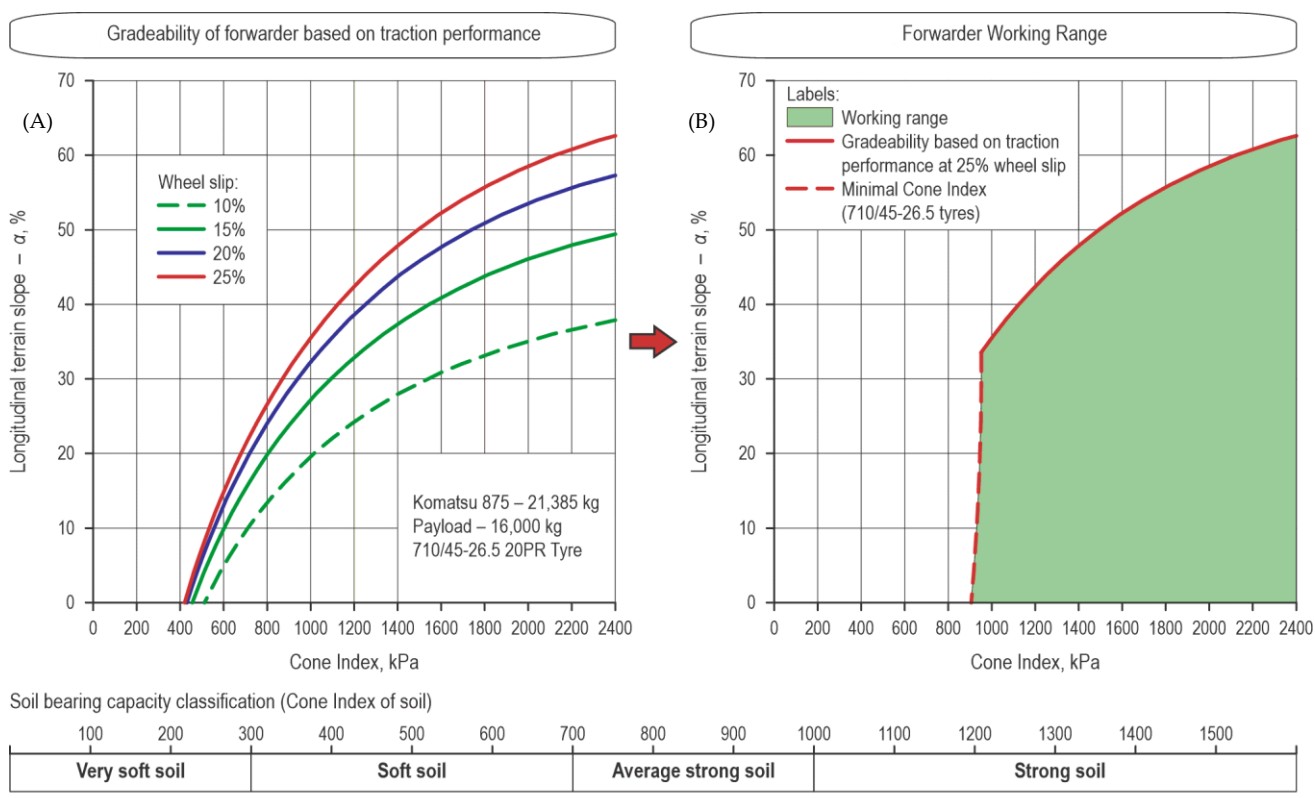

**Figure 7.** Gradeability (**A**) and working range (**B**) of Komatsu 875 forwarder [42].

The area of eco-efficient timber forwarding of the nominally loaded Komatsu 875 eight-wheel forwarder equipped with 710/45-26.5 tires is shown as the intersection of the curves of the limit slope of the forwarders' mobility at wheel slip of 25% and the minimal cone index (Figure 7B). Soil bearing capacity is calculated by the influence of the minimal cone index at 950 kPa and a limit slope of mobility at 33%. With a further increase of the cone index, the limit slope of forwarder mobility also increases 1 MPa → 36%, 1.2 MPa → 42%, 1.4 MPa → 48%, 1.6 MPa → 52%, 1.8 MPa → 56%, 2.0 MPa → 58%, 2.2 MPa → 61%, 2.4 MPa → 63%, 2.6 MPa → 64%.

## 4. Discussion

The presented model of the forwarder's mobility during timber forwarding uphill is based on the position of the centre of gravity of the unloaded forwarder, which is not mentioned by the ISO 13860 [36] standard, nor is it reported by manufacturers of forwarders. The vehicle's centre of gravity is an important constructive indicator, which greatly influences the traction characteristic and stability of the vehicle's mobility, and represents the point where the entire mass of the vehicle is concentrated [43]. Due to the lower driving speed of forest vehicles, if compared to road vehicles, the position of the centre of gravity of the forwarder is static, an essentially unchanging feature, which does not change dynamically during its acceleration, braking, or turning [35].

Due to the increasing use of forwarders on sloped terrain [12,13,44,45], the centre of gravity gains importance when modelling the distribution of axle loads, contact pressures, and the vehicles' mobility [4,41,46] to plan timber forwarding eco-efficiently.

Various methods are used to determine the position of the vehicle's centre of gravity: vertical hang [47], pendulum [48,49], lifting axle [37,50,51], and tilt table [52]. Determining the position of the centre of gravity of the Komatsu 875 eight-wheeled forwarder using the method of lifting the axle on the lifting platform and measuring the load on the forwarder wheels with portable scales indicated that the centre of gravity can be determined even without the expensive specialised measuring equipment (tilt table) that certification institutions usually have. Using the lifting platform ensured that all the wheels on the axle were in a horizontal position (Figure 3); in this way, the portable scales measured vertical loads, not resultant forces. The results indicated that the position of the centre of gravity of the forwarder also depends on the position of the hydraulic crane (crane in the position of driving or the extended position) during the measurement of the wheel load on the horizontal surface and sloped surface, which should be included in the normative documents. In addition to the above, the lifting platform proved to be suitable for measuring two additional indicators of forwarder mobility: (1) oscillation of the frame (Figure 8A) and (2) bogie axle assembly wheelbase angle (Figure 8B).

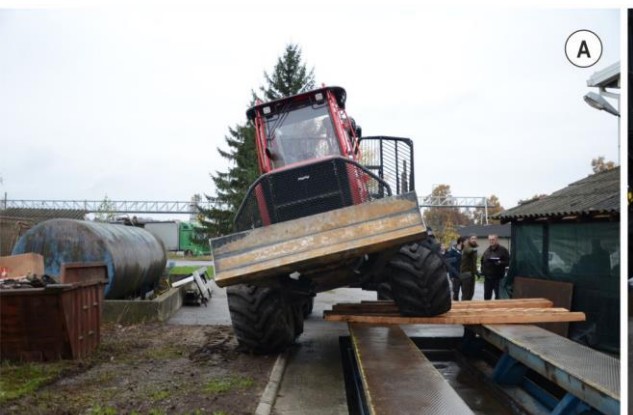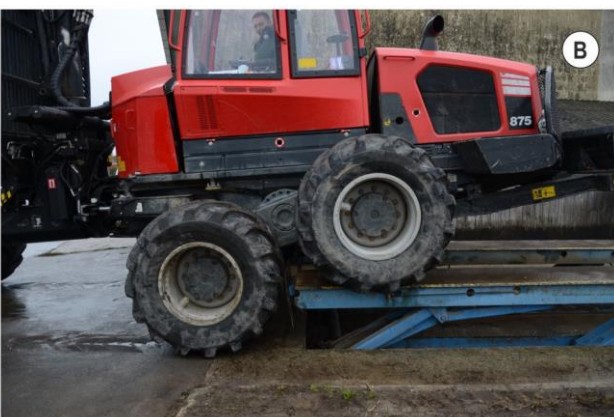

**Figure 8.** Measurement of frame oscillation (**A**) and bogie axle assembly wheelbase angle (**B**).

By knowing the position of the centre of gravity of the empty forwarder and the length of the transported timber that corresponds to the nominal load (assuming a full cross-section of the loading area) and by setting the equilibrium equations around the forwarder axles, it is possible to calculate the load on the forwarder's front and rear axles depending on the longitudinal terrain slope. The limitations of forwarder application, primarily developed for KWF's load distribution plan on a horizontal surface [20–22]: (1) the maximum load of loaded timber must not exceed the officially declared forwarder payload; (2) the maximum allowable load on the front axle; (3) the maximum allowable load of the rear axle must not be exceeded (whereby the sum of the load capacity of the tires per axle should be taken into account); and (4) the load on the front axle must not decrease below 20% of the weight of the loaded forwarder; these are also applicable for timber forwarding uphill with the note that they refer to adhesive axle loads that are perpendicular to the longitudinal terrain slope. The minimum load on the forwarder's front axle, which corresponds to 20% of the weight of the loaded forwarder, is a well-chosen criterion needed to maintain the forwarder's controllability. Sever [53] defines the same criterion for a cable skidder as the smallest ratio of adhesion load of the skidder's front and rear axle when skidding timber on sloped terrain ($G_1:G_2 > 1:3.5$). Other assumptions for calculating the mass of loaded timber (density of hardwoods of 1000 kg/m$^3$ and softwood of 700 kg/m$^3$, utilisation of the cross-section of the loading area of 70%), are easy to modify with regard to local conditions. Poršinsky et al. [19] researched the productivity of timber forwarding with regard to the correctness of timber scaling on the example of the forwarder Valmet 840.2. The authors determined the density of the common oak (*Quercus robur* L.) from 992 to 998 kg/m$^3$ of the gross volume of loaded timber, and the utilisation of the cross-section of

the loading space had the following values: 65% when loading logs and 4 m long firewood, 72% when loading 4 m long firewood, or 75% when loading logs.

The uniform distribution of axle loads in order to calculate the wheel load is the biggest limitation of the presented model because the front and rear wheels within the bogie axle are not absolutely equally loaded (Figure 4). However, without the wheel load calculated this way, it was impossible to simulate the gradeability of timber forwarding nor the nominal ground pressure. The case of non-uniform loading of the wheel on the bogie axle also occurs during the driving of the forwarder and is caused by the thrust force [54].

The highest nominal ground pressure of the reference (rear) wheels of the nominally loaded Komatsu 875 forwarder equipped with 710/45-26.5 tires of 132 kPa, and the highest value of the minimal cone index of 950 kPa indicated that the eco-efficient timber forwarding is possible only on strong soil. By equipping the forwarder Komatsu 875 with wider tires 800/40-26.5 (nominal ground pressure was 117 kPa, minimal cone index at 845 kPa) or with the use of high flotation tires 54×37-25 (nominal ground pressure was 94 kPa, minimal cone index at 674 kPa) satisfied all conditions of forwarder application on moderately strong soils. By additionally equipping the wheels of the front and rear bogie axles with tracks, the nominally loaded forwarder Komatsu 875 becomes eco-efficient even in conditions of soft soils [25].

The gradeability of the nominally loaded forwarder indicated that it significantly depends on the soil bearing capacity expressed by the cone index and the wheel slip. For the same values of cone index, at values of 10–15% of wheel slip, which corresponds to the highest tractive efficiency [32], the limiting slope of forwarder mobility has lower values, compared to wheel slip of 20–25% which corresponds to the environmental limitation and is defined as a beginning of erosion processes on sloping terrain [8]. An increase in wheel slip > 25% does not limit the thrust force on the vehicle wheel but instead leads to a reduction in vehicles' speed, a significant energy loss, and an increase in damage to the forest soil [55,56]. The applicability of the forwarder defined in this way should certainly be understood through the results of permanent soil cone index measurements during the year (on the example of district brown soils of hilly beech–fir forests), which indicate that, during the year, the soil cone index ranged from 800 kPa to 1200 kPa, except for the month of July when it reached values of 2200 kPa [57]. In cases of reduced soil bearing capacity as shown in the diagram in Figure 5B, the mobility of the forwarder on sloping terrain can be ensured by the winch-assist of the vehicle [15].

The impact of reducing the load (loaded timber) on the forwarders' gradeability was not considered due to the unfavourable impact on the productivity of timber forwarding [18,19].

## 5. Conclusions

Using the example of an eight-wheeled forwarder, a simulation model for assessing the mobility of forest vehicles during timber forwarding uphill was presented. The presented model enables the understanding of changes in vertical, horizontal, and traction forces during timber forwarding by a nominally loaded forwarder due to a wide range of changes in influencing factors: (1) slope of the terrain and (2) soil bearing capacity expressed by the cone index of the soil. By incorporating the criteria/limitations of timber forwarding from previous research, the model of forwarder mobility gains applicability as the theoretical approach brings the reality of timber forwarding. The importance of the presented model of mobility assessment is certainly reflected in the fact that it is possible, with satisfactory accuracy, to obtain essential knowledge related to the mobility of forest vehicles without long-term and expensive field research, the applicability of which is related to the researched forwarder and the conditions in which the measurements were made.

The presented model of forwarder mobility is based on easily measurable or available data of forest vehicles (mass, dimensions) but also on the centre of gravity, which is more challenging to measure and is not presented in the manufacturers' catalogues, but in this research was determined relatively easily.

For eco-efficient and safe timber forwarding, the presented forwarder mobility assessment model can be used by: (1) forestry experts as a tool for assessing the applicability of forest vehicles before purchase, i.e., planning mechanised timber extraction with the aim of increasing the harvesting system efficiency, and (2) forestry students during their education.

**Author Contributions:** T.P. and Z.B. conceived and designed the experiments; A.Đ. performed the experiments; Z.B. and Z.P. analyzed the data; T.P. contributed analysis tools; Z.B. and A.Đ. wrote the paper. All authors have read and agreed to the published version of the manuscript.

**Funding:** The research was carried out as a part of the project "Conservation of narrow-leaved ash stands (*Fraxinus angustifolia* Vahl) in the Republic of Croatia with an emphasis on harmful biotic factors" funded by the Ministry of Agriculture of the Republic of Croatia from the Funds for The Multifunctional Role of Forests.

**Data Availability Statement:** Not applicable.

**Conflicts of Interest:** The authors declare no conflict of interest.

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
