# Peer review of "Gradeability of a Forwarder Based on Traction Performance"

_forests, doi:10.3390/f14010103_

Round 1

Reviewer 1 Report

An interesting article on important problems of logistic processes during the planning of timber skidding on sloping areas. The results presented can be used utilitarianly. The authors did not avoid editorial errors in the text of the article. They have been highlighted and commented on in the attached file.

The methodological part should include an explanation of how to measure the index called "Cone index", as relevant to the developed simple method of determining acceptable operating parterms for safe forwarder operation on inclines.

According to the information recorded in lines 453-453, I encourage the authors to undertake research and prepare another article addressing the impact of reducing the load on vehicle axles on the efficiency of timber forwarding.

One of the bibliographic items in the bibliography list was not cited in the text of the article.

Author Response

REPLY TO REVIEWER 1

Thank you for all your comments.

An interesting article on important problems of logistic processes during the planning of timber skidding on sloping areas. The results presented can be used utilitarianly. The authors did not avoid editorial errors in the text of the article. They have been highlighted and commented on in the attached file.

The methodological part should include an explanation of how to measure the index called "Cone index", as relevant to the developed simple method of determining acceptable operating parterms for safe forwarder operation on inclines.

According to the information recorded in lines 453-453, I encourage the authors to undertake research and prepare another article addressing the impact of reducing the load on vehicle axles on the efficiency of timber forwarding.

One of the bibliographic items in the bibliography list was not cited in the text of the article.

All the changes have been made in the new document and are highlighted red.

Cone index was additionally explained.

The bibliographic which you are mentioning is actually in Fig. 7

Reviewer 2 Report

The authors consider the static center of gravity unjustified. However, in real operating conditions, it is necessary to study the dynamics of the process and determine the center of gravity only in the dynamics.

The authors in fig. 2a, fig. 2 (below), fig. 3 (bottom right photo), fig. 8a, provide a front dump loader, and in fig. 1, fig. 2b, fig. 4, fig. 8b no longer has a front loader dump.

A fundamental unfounded assumption is the consideration of a point contact. Although world engineering is already considering the conditions of the elastic-deformed state of the elastic wheel. The authors avoid the phenomenon of deformation of the elastic wheel, and the spot of contact with a variable area.

Fig. 4 – the authors unjustifiably neglect the kinematics of the manipulator placement and its weight indicators.

Author Response

REPLY TO REVIEWER 2

Thank you for all your comments.

The authors consider the static center of gravity unjustified. However, in real operating conditions, it is necessary to study the dynamics of the process and determine the center of gravity only in the dynamics.

The authors in fig. 2a, fig. 2 (below), fig. 3 (bottom right photo), fig. 8a, provide a front dump loader, and in fig. 1, fig. 2b, fig. 4, fig. 8b no longer has a front loader dump.

A fundamental unfounded assumption is the consideration of a point contact. Although world engineering is already considering the conditions of the elastic-deformed state of the elastic wheel. The authors avoid the phenomenon of deformation of the elastic wheel, and the spot of contact with a variable area.

Fig. 4 – the authors unjustifiably neglect the kinematics of the manipulator placement and its weight indicators.

All the changes have been made in the new document and are highlighted red.

As it is mentioned in the Disscusion part of the MS due to the lower driving speed of forest vehicles, if compared to road vehicles, the position of the centre of gravity of the forwarder is a static, an essentially unchanging feature, which does not change dynamically during its acceleration, braking or turning [35]. And this can be additionally explained by the use of a forwarder in thinning operations where: „When driving on the road, the forwarders had an average speed of 71.6 m min-1 empty and 75.7 m min-1 loaded. When driving in the stand, the average speed was 56.9 m min-1 empty and 52.2 m min-1 loaded.“ By Hildt, E., Leszczuk, A., Donagh, P. M., & Schlichter, T. (2020). Time consumption analysis of forwarder activities in thinning. Croatian Journal of Forest Engineering, which was not cited in the uploaded MS.

The stacking blade (as a presumption, that is what you mean by a "front loader dump") was always in the same position during the measurements  (even though it is not visible in some photos, that is only because of the angle of the recording). Regarding the lack of the stacking blade on schematic figures, the reason for that is that other vehicles' dimensions (characteristics) wanted to be highlighted and shown, which would be more difficult with the added stacking blade.

The placement of the crane (if that is what you mean by the „manipulator placement“) was not neglected since the difference in the position of the centre of gravity is the result of moving part of the mass of the hydraulic crane's inner, outer and telescope boom towards the rear axle of the vehicle, as well as positioning the crane in the plane of the stake height.

Round 2

Reviewer 2 Report

Formula (11) has an incorrect dimension (kg %) = (m2 % kg/m3 m).

Author Response

REPLY TO REVIEWER 2, round 2

Thank you for your comment.

Formula (11) has an incorrect dimension (kg %) = (m2 % kg/m3 m).

The formula has been corrected and is now marked in green.